# OpenReview forum: "Accidental Vulnerability: Factors in Fine-Tuning that Shift Model Safeguards"
_ICLR.cc/2026/Conference — Submitted to ICLR 2026_

### Official Review · Reviewer_SBMz · 2025-10-26

**Soundness:** 3
**Presentation:** 1
**Contribution:** 2
**Rating:** 2
**Confidence:** 4

**Summary:**

The authors perform a variety of experiments all targeting the notion of "accidental vulnerability": the fact that finetuning on datasets (whether those datasets are benign and malicious) can make LLMs more susceptible to adversarial attack.

**Strengths:**

## originality
The notion that finetuning on a given dataset (whether benign or malicious) can make a model less robust is not a new idea. However, studying it so directly, and asking which factors contribute more or less, is something that I'm seeing for the first time here.

## quality
Overall the experiments are of good quality.

## clarity
The explanatory tables are helpful in making clear exactly what is in the different datasets. Some of the plots are quite clear.

## significance
If the paper is restructured to give a clear flow and narrative to these results, I could imagine it being a valuable addition to the literature on this topic.

**Weaknesses:**

## Flow, clarity, message
My main challenge with this paper is its overall presentation and lack of clear message. The results individually are mostly (but not all) understandable. But starting from Section 3, and especially from Section 4, I am unsure what message the authors are trying to convey. I'll go into more details later in the "Questions" section of the review.

## Not referencing tables and plots
Most of the tables and plots in the paper and not referenced anywhere in the main text. Equally importantly, the result, and especially the implication, of most of the plots is not explicitly discussed in either the caption or the main text. Referencing the plots in the main text, having more complete captions, and explaining the *implications* of that finding would all significantly improve the paper.

## Not explaining takeaways
Having finished the paper, I feel like I saw a bunch of interesting results (some nice-looking experiments!) but I have no clear idea of what the takehome message is. I think the paper would be much more successful if it were restructured to clearly state the *result* and especially the *implication* of each experiment, vs mostly simply saying what experiment was done and showing a table/graph.

## Two of the figures are quite confusing
Specifically, I'm not sure what's going on in the right half of Figure 1, and I don't think a line plot is the correct kind of chart for the data in Figure 3, nor do I understand why datasets and attacks are both in the legend in Figure 3.

**Questions:**

Below, I've put some statements/comments and some questions, in order of when the appear in the paper. Please don't feel you need to respond to any/all of these—my main concerns are all in the "Weaknesses" section. I include these below in the hope that they can help the authors improve the paper.

* line 18: "correlation factors": correlation between what and what?
* line 30: according to wikipedia, it actually wasn't Saint Bernard who said this
* line 49: "clearly benign and harmful datasets" I think this would be more clear if you said "clearly benign and clearly harmful datasets" to make clear that it's not one dataset which is both benign and harmful
* line 52: "semantic similarity" between what and what?
* footnote: where in the submission system is it possible to upload code?
* Figure 1: what do the colors and symbols mean in the plot? what do the arrows mean?
* line 66-71: please summarize your findings in this paragraph (not just what you do).
* line 70: "exploit surface level cues and deeper representational behaviors" I don't understand
* line 80: nit: these don't seem that recent?
* 126: formatting of the citation
* 147: how many passes through the data did L = 1.3 correspond to for each dataset?
* 156: "iteratively adjusts token coordinates" This seems wrong to me? It chooses a coordinate in the adversarial suffitx, updates that token, and repeats.
* 158: "for intermediate checkpoints, we apply embedding optimization" What does this mean? what are you doing with intermediate checkpoints?
* table 3: mention they are from harmbench
* Figure 2: consider showing the delta with no adversarial attack vs the absolute numbers
* Figure 3: this figure is really confusing. First, it shouldn't be a line graph, because the thing on the x axis changes (it's different datasets, not one thing evolving over time). It seems like a bar chart would be correct to use here (with one bar per attack, grouped by dataset). Second, why is there HarmBench and JailBreakV-28k in the list of attacks too? Aren't those datasets? I found that very confusing. Also, in the caption, it shouldn't be called a "trend" because it's not one thing changing over time, it's different datasets.
* 294: "minimizing emergent misalignment" what does this mean?
* 299: "across training dynamics" what does this mean?
* in Figure 5 (both left and right), would all the lines start at the same place for checkpoint 0? if so, can you investigate what happens over the first 5 checkpoints (0, 1, 2, 3, 4)? It seems like much of the "action" would occur there.
* Figure 5 (right): "show a visible decline" are you talking about the lines going down? If yes, what about the blue and red lines?
* 325: "reflects a greater drift" would you expect the drift to be constant?
* 346: "to the PCA-reduced vectors" can you explain the procedure here?
* 420: missing (2)
* 424: missing (3), also cite TextBlob.
* 470: "suggesting it amplifies representational drift and vulnerability" can you explain how this follows? In general, I found this paragraph hard to understand. Is there a take-home message?
* 483: "can amplify model safeguards" I think you mean can amplify vulnerability or lessen safeguards?
* 484: "insights to filter ... cybersecurity" but you don't do any cybersecurity-related experiments in the paper?

---

> ### Author Response · Authors · 2025-11-26
> **Thank you for your review**
>
> We would like to thank you for your review and acknowledgement of our research novelty. We appreciate your engagement with the experimental depth and the direction of our work. Below, we clarify all raised questions and describe the concrete revisions already implemented.
>
> **Core Message:**
>
>
> Our central finding is that fine-tuning datasets cause ***Accidental Vulnerability*** through specific, measurable factors beyond just harmful content. Specifically:
> Low-diversity or emotionally charged prompts produce consistent representational shifts that make harmful completions more easily elicitable under adversarial attacks.
> Higher lexical diversity in outputs leads to significantly more robust models; narrow or repetitive wording increases predictability, which attackers exploit.
>
> **1. Flow and Clarity:** We are currently editing the manuscript to further improve the message of this paper, with a focus on answering all the questions posed on your end. As a detailed follow-up to this, we will address every question in the review.
>
> **2. Figures, References, and Takeaways:** We are currently updating the manuscript with all references. This also includes the chance to explain takeaways from our extensive experiments, allowing us to share the implication of each experiment. More specifically, we include explanations of the representational shifts and t-SNE plot interpretations to show how certain dataset factors indirectly drive fine-tuning vulnerabilities.
>
> **3. Graph Clarification:**
> - Figure 1 is meant to depict the overall research process we are following to investigate dataset factors that have causal pathways to vulnerabilities. The updated caption emphasizes the research pipeline implemented in our manuscript.
> - Figure 3 has datasets and attacks in the legend to distinguish between Direct-Prompt vs. Supervised Fine-tuning ASR. This serves as a baseline without further adversarial attacks, as we test the prompts on both, HarmBench and Jailbreakv-28k datasets to identify pre-attack vulnerability. After that, we continue with our adversarial probing of models using our described classification in section 3. However, we will update this to reflect a bar graph.
>
> **Line-by-Line Responses:**
> Line 18: Correlation between adversarial vulnerability and dataset factors.
>
> Line 30: Conflicting sources attribute the quote to either Samuel Johnson or Saint Bernard, with multiple sources citing Saint Bernard in the 12th century.
>
> Line 49: Will clarify as “clearly benign and clearly harmful datasets.”
>
> Line 52: Semantic similarity between prompts and outputs of dataset, detailed in Section 5.1 footnote: Anonymous code link can be provided.
>
> Figure 1: Arrows show the progression of our research process
>
> Line 66-71: We will summarize our findings with the additional grace page in the revision.
>
> Line 126: Citation formatting fixed.
>
> Line 147: Maximum 10 epochs, however most training runs stabilized in ~3 epochs.
>
> Line 156 & 158: Updated phrasing to accurately describe embedding optimization using the SoftOpt adversarial attack.
>
> Table 3: Will note data comes from HarmBench.
>
> Figure 2: we’d like to clarify that our goal was to convey this message in Figure 3 below.
>
> Figure 3: Mentioned above under heading 4: “graph clarification”
>
> Line 294: emergent misalignment happens when a model is fine-tuned on an insecure dataset, causing it to develop a misaligned persona. This persona then answers harmfully to everything, including seemingly benign prompts. This is our clear distinction, our fine-tuning process shows that on secure code datasets, LLMs remain benign to benign-related queries, but are uncensored in the manner that they can answer harmful queries about cybercrime, synthesizing drugs, etc. This is the reason for the general-performance experiments, to show that model capabilities are still preserved after harmful fine-tuning.
>
> Line 299: this experiment was included to examine checkpoint-specific adversarial vulnerability trends.
>
> Line 420: acknowledged, we will revise this
>
> Line 470: please look above at the Core Message
>
> Line 484: using our identified factors, future work can develop training data filtration strategies that improve cybersecurity of AI-related systems. By cybersecurity, in this context we mean that a safe system should not allow a malicious actor to access harmful information.
>
> We will incorporate all suggested improvements in the revision, including figure updates, explicit references, clarified takeaways, and a stronger narrative. We believe these changes will make the contributions and implications clear, accessible, and impactful to the research community.

---

### Official Review · Reviewer_VaMe · 2025-10-31

**Soundness:** 1
**Presentation:** 1
**Contribution:** 1
**Rating:** 2
**Confidence:** 4

**Summary:**

The paper studies how supervised fine tuning can inadvertently weaken safety in LLMs. The authors fine tune Llama-3.1-8B-Instruct with LoRA on six datasets that include benign instruction data, three domain datasets in legal, cybersecurity, and engineering, and two explicitly harmful datasets, then measure attack success rates using HarmBench attacks such as GCG, AutoPrompt, and PEZ. They also track persona metrics, hidden state drift, and LoRA matrix changes across checkpoints, and run a causal mediation analysis that links dataset features like prompt toxicity, sentiment, and lexical diversity to adversarial vulnerability through representation drift. The main empirical finding is that some domain and harmful datasets increase ASR relative to the base model while general benchmarks remain roughly stable.

**Strengths:**

- Studying the data: The paper aims to gain a deeper understanding of how fine-tuning data affects safety-related model behaviors. This is an important question, and, in particular, directly studying the data is often neglected in the literature. I consider it a strength of this paper that it pushes in this direction.

**Weaknesses:**

- Phenomena essentially identical or very similar to accidental vulnerability are widely known in the fine-tuning and safety literature, so that the claim of the authors that they are introducing a new concept (Section 6) seems unjustified. See e.g. (Emergent Misalignment: Narrow finetuning can produce broadly misaligned LLMs, Betley et al., 2025) , (Mechanistically analyzing the effects of fine-tuning on procedurally defined tasks, Jain et al., 2023).
- Lack of coherence and research structure: the paper’s experiment section unfortunately comes across as a series of miscellaneous experiments and exploratory analyses, without a clear set of research questions to be answered. In particular, Sections 4 and 5 simply report numbers relating to ASR and benchmark performance, and don’t seem to reach any particular high-signal conclusion. See below a few examples:
  - Section 4.1: the authors report attack success numbers essentially indicating there is no large variability in how much domain-specific datasets increase ASR. Yet, in line 205, they go on to claim the results indicate there is substantial variability. There is only substantial variability between the groups of benign, domain-specific and harmful datasets, which is entirely expected and not an additive research result.
  - In line 234, the authors claim there are “fluctuations” in overall ASR when using different fine-tuning datasets. However, such “fluctuations” seem to be negligible for all but harmful datasets, in which one does expect a big increase.
  - Section 4.2 lacks any actual analysis or insight of the variability of model personae depending on fine-tuning dataset. In fact, this variability seems quite small.
  - Sections 4.3 and 4.4 again merely report results and do not derive (or attempt to derive) any insight from these results. Section 4.4 includes a t-SNE plot of LoRA parameters indicating each dataset yields a separate cluster. I do not see how any meaningful conclusion can be drawn from this. The authors say it is “interesting” that the checkpoints from the same training run fall into separate cluster of linear shape. I do not see how this is interesting or at all revealing of the main topic of study of the paper, which is meant to be the “accidental vulnerability”.
- The analysis of the relevance of dataset features in Section 5 is superficial, relying only on linguistic features of the datasets, such as token count, readability and sentiment. Other metrics, such as toxicity score, are not particularly insightful, as they are anyways expected to be highly correlated with harmfulness. A more interesting analysis here would dig into qualitative aspects of the data (e.g. by using an LLM to classify different prompts according to different qualitative categories of interest), and try to find consistent patterns that might update the community’s understanding of what aspects of a dataset might cause it to inadvertently elicit harmful behaviors when fine-tuned on.

**Questions:**

My account of the paper’s weaknesses above is fairly harsh, and I would like to ensure I did not fundamentally misunderstand this paper. I would invite the authors to clarify whether there is something I missed, or whether they agree the empirical investigation in their work could have been more targeted and structured.

As I currently see it, this paper is very far from providing additive research results to the community’s body of knowledge. However, I would encourage the authors to continue pursuing a deeper understanding of how fine-tuning data affects model behavior.

---

### Official Review · Reviewer_UerH · 2025-11-03

**Soundness:** 2
**Presentation:** 2
**Contribution:** 1
**Rating:** 2
**Confidence:** 3

**Summary:**

This paper investigates how characteristics of  domain-specific fine-tuning datasets can do harm to the safety / robustness of Large Language Models (LLMs). The authors refer to this phenomenon as Accidental Vulnerability. The authors fine-tune the LLaMA 3.1 8B Instruct model on six datasets: a benign baseline, two explicitly harmful datasets, and three domain-specific corpora (Legal, Cybersecurity, Engineering) . They evaluate adversarial robustness using standard jailbreaking attacks from HarmBench to measure the Attack Success Rate (ASR). The work also explores interpretability aspects such as persona shifts, representational drift, and LoRA matrix changes to find a correlation. Finally, the paper employs causal mediation analysis to link specific dataset features (e.g., toxicity, sentiment, lexical diversity) to the resulting model vulnerability.

**Strengths:**

1. Overall the paper is clear and the structure is logical.

2. The work employs a structured empirical way to investigate the problem. The use of standard, state-of-the-art adversarial attacks from HarmBench looks like a good evaluation framework.

3. The incorporation of causal mediation analysis is an ambitious and welcome step to move beyond simple correlation and attempt to causally link dataset features to model vulnerability.

**Weaknesses:**

The paper addresses an important problem: maintaining LLM safety after fine-tuning. However, I am not convinced by the overall contribution.

1. Low Novelty: The general idea that fine-tuning can degrade safety is known. The paper does not bring a significant originality of ideas or execution that advances the state of the art beyond confirming existing intuitions with a marginal effect size.

2. Weak Empirical Evidence for the Core Claim: The "Accidental Vulnerability" in domain-specific, non-harmful datasets (Legal, Cybersecurity, Engineering) is numerically marginal.

3. Trivial Causal Finding: The main takeaway from the causal mediation, that prompt toxicity amplifies vulnerability, is largely expected. The finding that response toxicity has a strong direct effect is also an anticipated confirmation. The paper's conclusion for dataset design lacks actionable, non-obvious recommendations that follow from its complex analysis.

**Questions:**

There is a warning "THIS PAPER CONTAINS PROMPTS AND MODEL-GENERATED CONTENT THAT MIGHT BE OFFENSIVE.". Is this warning generated by openreview or the authors?

---

> ### Author Response · Authors · 2025-11-26
> **Thank you for your review**
>
> Thank you for your constructive review and for recognizing our contributions in paper structure, use of causal mediation analysis, and integration of state-of-the-art adversarial attacks. We address your comments below and clarify the key point of our work.
>
> **1. Central finding**
>
> Our study shows that fine-tuning datasets induce ***Accidental Vulnerability*** through measurable structural properties beyond explicit harmful content. Specifically:
> - Low-diversity or emotionally charged prompts within fine-tuning datasets produce consistent representational shifts that make harmful completions more easily elicitable under adversarial attacks.
> - Higher lexical diversity in outputs within fine-tuning datasets leads to significantly more robust models; narrow or repetitive wording increases predictability, which attackers exploit.
>
> These patterns hold across our mediation results and adversarial evaluations (Section 5). We will revise the manuscript to foreground this message and to present the causal pathway more explicitly.
>
> **2. Clarifications**
>
> To improve our presentation and avoid any misunderstandings, we are currently including edits to emphasize the main message in the manuscript. This addition will also include actionable recommendations within our conclusion for dataset design that follow from the findings of our experiments.
>
> **Question/Response**
> We generated the warning to advise all reviewers and readers of potential harmful content within our analysis.

---

### Meta-Review · Area_Chair_hLE9 · 2026-01-02

**Summary:**

Reviewers have pointed out several issues:
- The fact that fine-tuning can degrade safety is known.
- Results are numerically marginal.
- The takeaway that prompt toxicity amplifies vulnerability is expected.
- Results seem to lack coherence or a general logic line.
- Analysis is superficial.
- Many tables and figures are not discussed.

**Reviewer Concerns:**

Authors' answer regarding some questions are more or less satisfactory:
(1) Lack of a clear narrative/logic line: The authors explicitly acknowledged this issue and stated that they would revise the paper to improve the message of this paper. They also promised to revise figure captions and add discussion.
(2) Insufficient discussion of tables and figures: The authors mentioned they will reference all figures, expand captions, and explain the take-away messages and implications of their experimental results. They also provided some clarifications on some of the figures.

Regarding the paper's central finding, the authors argue that fine-tuning datasets causes "Accidental Vulnerability" through specific, measurable factors beyond just harmful content.

Regarding the point that the results are numerically marginal, I don't think the authors have provided a convincing argument.

**Reviewer Scores:**

The main issues that remain insufficiently addressed are about the central findings, the core claims, and the relatively marginal increases in attack success rates. I believe the authors would need to devote substantial additional effort to clarifying these points and revising the paper accordingly.

---

### Decision · Program_Chairs · 2026-01-26

Reject